# Analysis of vibronic coupling in a 4f molecular magnet with FIRMS

Jon G. C. Kragskow [1,5], Jonathan Marbey[2,3,5], Christian D. Buch [4], Joscha Nehrkorn[2], Mykhaylo Ozerov [2], Stergios Piligkos [4✉], Stephen Hill [2,3✉] & Nicholas F. Chilton [1✉]

Vibronic coupling, the interaction between molecular vibrations and electronic states, is a fundamental effect that profoundly affects chemical processes. In the case of molecular magnetic materials, vibronic, or spin-phonon, coupling leads to magnetic relaxation, which equates to loss of magnetic memory and loss of phase coherence in molecular magnets and qubits, respectively. The study of vibronic coupling is challenging, and most experimental evidence is indirect. Here we employ far-infrared magnetospectroscopy to directly probe vibronic transitions in [Yb(trensal)] (where $H_3$trensal = 2,2,2-tris(salicylideneimino)tri-methylamine). We find intense signals near electronic states, which we show arise due to an "envelope effect" in the vibronic coupling Hamiltonian, which we calculate fully ab initio to simulate the spectra. We subsequently show that vibronic coupling is strongest for vibrational modes that simultaneously distort the first coordination sphere and break the $C_3$ symmetry of the molecule. With this knowledge, vibrational modes could be identified and engineered to shift their energy towards or away from particular electronic states to alter their impact. Hence, these findings provide new insights towards developing general guidelines for the control of vibronic coupling in molecules.

[1] Department of Chemistry, School of Natural Sciences, University of Manchester, Oxford Road, Manchester M13 9PL, UK. [2] National High Magnetic Field Laboratory, Tallahassee, FL 32310, USA. [3] Department of Physics, Florida State University, Tallahassee, FL 32306, USA. [4] Department of Chemistry, University of Copenhagen, DK-2100 Copenhagen, Denmark. [5]These authors contributed equally: Jon G. C. Kragskow, Jonathan Marbey. ✉email: piligkos@chem.ku.dk; shill@magnet.fsu.edu; nicholas.chilton@manchester.ac.uk

Vibronic coupling is pervasive—all materials vibrate and have electronic states—and its impact is crucial in many settings. For example, it is thought to be central in photosynthesis[1,2] and in light-harvesting proteins[3], but, more generally, it is implicated in enantioselective catalysis[4] and luminescent materials[5], and is pivotal in the operation of molecular qubits[6–10] and single-molecule magnets[11]. Synthetic chemists have made extensive strides in controlling vibronic coupling through judicious molecular design[12], but the community at large is far from general design guidelines to control such effects. A key roadblock to progress is obtaining direct evidence of vibronic coupling: conventional experiments that probe magnetic relaxation and quantum phase coherence are only indirectly sensitive to the effects of vibronic coupling[6,8,11,13], while direct measurements such as ultrafast[12,14] or infrared (IR)[15–17] spectroscopies are rare.

In the context of molecular magnetism, vibronic coupling occurs in the presence of any molecular vibration that modulates terms in the Hamiltonian associated with a magnetic centre (e.g. the crystal field (CF) potential). Such couplings are directly related to spin relaxation phenomena, and most commonly associated with spin-lattice relaxation. Though these effects have been known since the work of van Vleck and Orbach[18,19], their specific relevance to single-molecule magnets and molecular qubits has only more recently been subject to intense study. Despite this recent surge in interest, few examples exist where the vibronic coupling has been characterised in detail[15–17,20]. To this end, here we perform far-IR magnetospectroscopy (FIRMS)[21,22] measurements on the Yb[III] qubit [Yb(trensal)] (1, where H$_3$trensal = 2,2,2-tris(salicylideneimino)trimethylamine, Fig. 1) to directly probe the vibronic coupling in this molecule, and develop ab initio simulations of the FIRMS map to elucidate the origins of the vibronic transitions. This molecule was chosen due to its extensive existing magnetic and spectroscopic characterisation: its electronic states and CF splitting has been studied with luminescence spectroscopy[13,23]; it is known that a two-phonon Raman process dominates the spin-lattice relaxation at low temperature;[13] and it shows long enough phase coherence times such that Rabi oscillations can be driven for nuclear-spin-specific transitions[24,25]. A FIRMS map is obtained from a series of far-IR spectra collected in varying magnetic fields, which is then normalised to remove field-independent signals corresponding to purely vibrational modes. Thus, a FIRMS map highlights purely electronic transitions (due to the magnetic dipole selection rule, $\Delta m_J = \pm 1$, as employed for instance in refs. [15,26,27]) and mixed

vibronic transitions which involve a simultaneous change in both electronic and vibrational states due to absorption of an IR photon (i.e. simultaneous electric and magnetic dipole transitions, where the selection rule for the former is $\Delta n = \pm 1$ for vibrational modes); this is distinct from transitions between electronic states induced by absorption of phonons, which are the origin of magnetic relaxation in single-molecule magnets and a contributor to quantum decoherence in molecular qubits. Nonetheless, we can learn a great deal about these latter effects from measurement and simulation of vibronic coupling.

Complex 1 has C$_3$ point symmetry and crystallises in the P$\bar{3}$c1 space group. Yb[III] has a 4$f^{13}$ ground configuration which is split into the ground $^2F_{7/2}$ and excited $^2F_{5/2}$ multiplets by spin–orbit coupling (Fig. 2, inset), which are then further split by the CF of the molecule; in the absence of a magnetic field all states are doubly degenerate owing to Kramers theorem[28]. Some of us have previously reported near-IR absorption and luminescence measurements of 1 in a diamagnetic host [Yb$_{0.07}$Lu$_{0.93}$(trensal)] (1') and have experimentally determined the CF splitting of both spin–orbit multiplets. Fitting the magnetic susceptibility, magnetisation, and optical data simultaneously with a CF Hamiltonian (Supplementary Tables 1 and 2), yields effective g-values for the ground doublet which match those from electron paramagnetic resonance (EPR) spectroscopy[24]. This reveals considerable axial and trigonal contributions to the CF, where nearly all states are mixtures of $m_J$ functions, except for the 3rd Kramers doublet (KD) which comprises the pure $m_J = \pm 3/2$ states as these cannot mix with other $m_J$ states in C$_3$ symmetry. Additional peaks are found in the luminescence spectrum of 1' which do not correspond to CF energy levels of the $^2F_{7/2}$ multiplet (Fig. 2, e.g., peaks 2a and 2b); these were attributed to "vibrational sidebands" in the original paper[13], but the true nature of these features was unknown. Here, we find that vibronic transitions appear near CF states due to a hitherto undescribed "envelope effect", and that vibronic coupling is strongest for vibrational

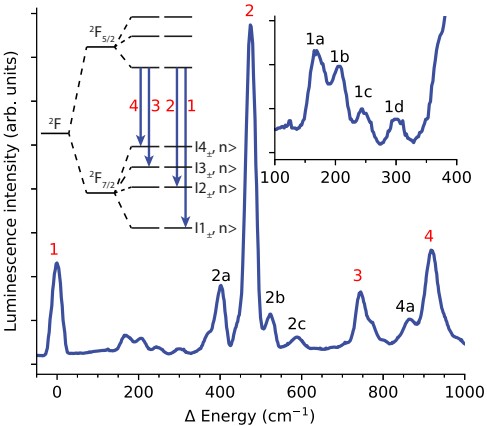

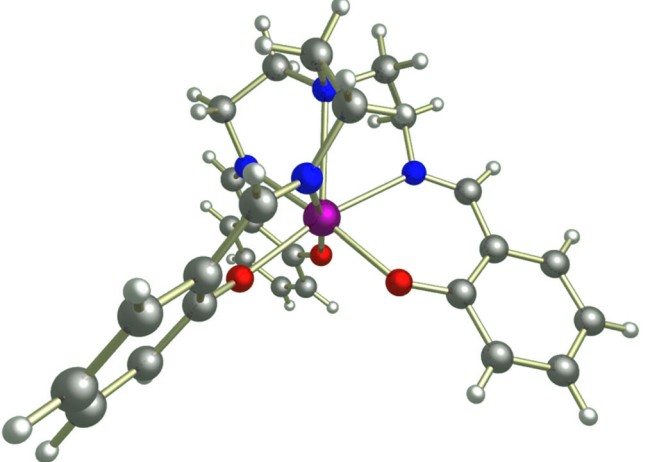

**Fig. 1 Structure of [Yb(trensal)] (1) viewed perpendicular to the C$_3$ axis.** Hydrogen = white, carbon = grey, nitrogen = blue, oxygen = red, ytterbium = purple.

**Fig. 2 Measurement and assignment of low-lying electronic states in [Yb(trensal)].** Experimental luminescence (emission) spectrum of [Yb$_{0.07}$Lu$_{0.93}$(trensal)] (1') at 5 K[13]. Transitions are from the lowest KD of the excited $^2F_{5/2}$ spin–orbit multiplet to the different KDs of the ground $^2F_{7/2}$ multiplet (inset left; not to scale) and occur in the near-IR around 980 nm[13,60]. The spectrum is plotted as energy differences with respect to the zero-phonon line of the ground KD (peak 1), thus the spectrum is reversed compared to a conventional emission spectrum. The energies of the four KDs of the $^2F_{7/2}$ multiplet, as determined from the spectrum relative to the ground KD at 0 cm$^{-1}$ (1) are: 474 cm$^{-1}$ (2), 745 cm$^{-1}$ (3) and 920 cm$^{-1}$ (4). Additional features are at 169 cm$^{-1}$ (1a), 207 cm$^{-1}$ (1b), 247 cm$^{-1}$ (1c), 302 cm$^{-1}$ (1d), 403 cm$^{-1}$ (2a) and 524 cm$^{-1}$ (2b), 588 cm$^{-1}$ (2c), 864 cm$^{-1}$ (4a).

modes that distort the first coordination sphere of Yb$^{III}$ as well as breaking the $C_3$ point symmetry. Such findings are paramount in unravelling the complex nature of vibronic coupling and for developing future molecular design criteria to deliver control of this phenomenon.

## Results and discussion

**Ab initio electronic structure**. Using the structure from X-ray diffraction (XRD), complete active space self-consistent field calculations with perturbative corrections for dynamic correlation and spin–orbit coupling (CASSCF-CASPT2-SO; see the "Methods" section) are in excellent agreement with the experimentally determined CF energies, where the first excited state is only ~40 cm$^{-1}$ lower than experiment (Supplementary Fig. 1, Supplementary Tables 2 and 3). The composition of the ground KD is very similar to the experimental CF model and to EPR data ($g_{\parallel,\text{calc}} = 4.98$ and $g_{\perp,\text{calc}} = 2.69$, cf., $g_{\parallel,\text{exp}} = 4.29$ and $g_{\perp,\text{exp}} = 2.90$)[24]. Comparing these results with those obtained from CASSCF alone shows the importance of dynamic correlation in the electronic structure of **1**, as both the $g$-values and CF energies are markedly closer to experiment (Supplementary Table 4). Optimisation of the structure of **1** using density-functional theory (DFT, see the "Methods" section) yields the structure **1$_{opt}$** (Supplementary Table 5), which shows only minor structural changes (root mean squared deviation of 0.127 Å compared to **1**). The vibrational modes of **1$_{opt}$** are classified as A (singly degenerate) or E (doubly degenerate) irreducible representations of the $C_3$ point group (Supplementary Table 6), and we find good agreement between the calculated vibrational energies and the experimental Fourier transform IR (FTIR) spectrum in zero-field (Supplementary Fig. 2). CASSCF-CASPT2-SO calculations on **1$_{opt}$** give a slightly poorer agreement with the overall CF splitting from the experimental spectrum (Supplementary Fig. 1), though the first excited state is now only ~10 cm$^{-1}$ lower than the experimental value and the ground state $g$-values remain practically unchanged (Supplementary Table 7). The considerable impact of small structural changes on the electronic states of **1** (excited states shift by ~30, ~70 and ~100 cm$^{-1}$, respectively) indicates that the electronic structure of **1** is highly susceptible to molecular distortion, providing a physical basis for significant vibronic coupling found for this molecule.

**FIRMS map and "toy model" Hamiltonian**. A FIRMS map highlights vibronic transitions driven by IR photons with energy $h\upsilon = \Delta_e \pm \Delta_v$, where $\Delta_e$ is the difference in electronic energy and $\Delta_v$ is the difference in vibrational energy. The intensity of a vibronic transition in a FIRMS map is related to the intensities of both the pure vibrational and the pure electronic transitions, but also the strength of vibronic coupling between the states involved. The FIRMS map for **1** (Fig. 3a, b) reproduces the vibronic side-bands observed in luminescence measurements (Fig. 2), and reveals evidence of their movement (along with several other features) as a function of applied magnetic field. While the zero-field FTIR spectrum of **1** shows vibrational modes ranging from 0 to 900 cm$^{-1}$, in good agreement with our DFT calculations (Supplementary Fig. 2), the FIRMS map shows far fewer field-dependent signals that appear in bands from 370–550 cm$^{-1}$ and 740–815 cm$^{-1}$ (Fig. 3a and Supplementary Fig. 7b) near the energies of the electronic doublets in **1** (474 and 745 cm$^{-1}$). Interestingly, the spectrum shows field-dependent vibronic signals below the energy of the first excited KD (i.e., 370–474 cm$^{-1}$), which mainly arise from very low-energy intra-KD electronic transitions (e.g. microwave energy range, EPR-type transitions) coupled to vibrational excitations near the observed transition energy (hot vibrational transitions are very unlikely at 4.2 K, see below and Supplementary Fig. 4). Given this, it is odd that vibronic transitions are not observed in other ranges with significant

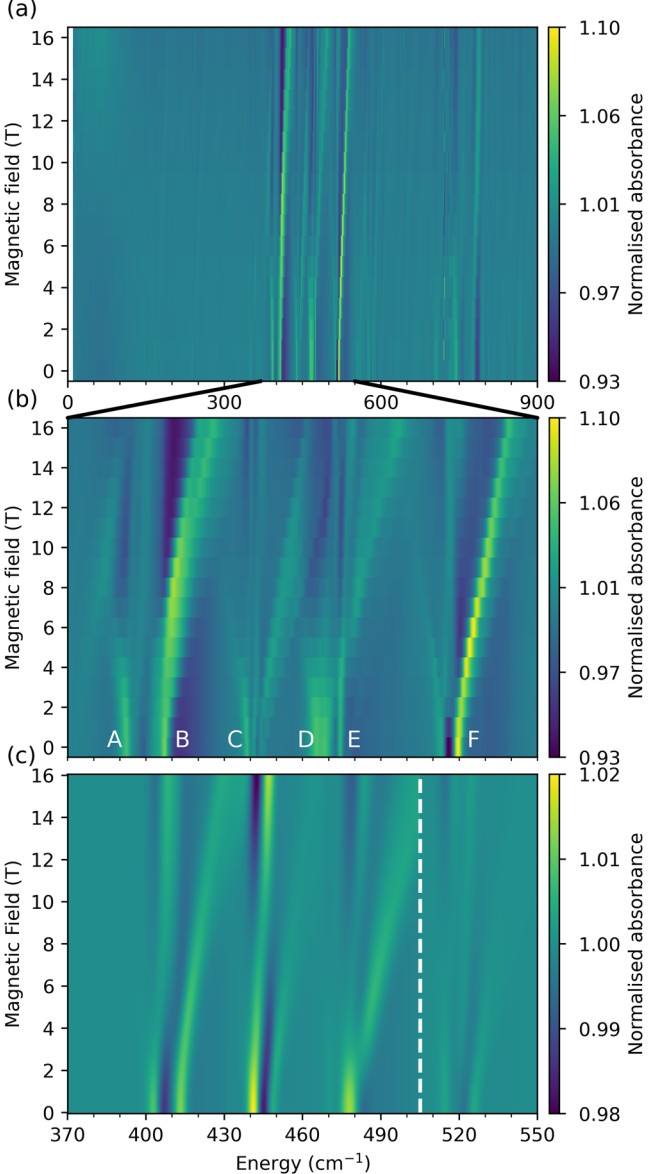

**Fig. 3 Experimental and simulated FIRMS maps for 1.** Experimental FIRMS map measured at 4.2 K in the range **a** 0–900 cm$^{-1}$ and **b** 370–550 cm$^{-1}$ (field dependent signals are labelled as **A–F**). **c** Composite ab initio simulated FIRMS map for signals **A–F** using CASPT2-SO equilibrium CF parameters with experimental CF energies, CASSCF-SO vibronic couplings and ab initio transition intensities where the vibrational transition probabilities have been scaled according to Fig. S3, and the electronic transition probabilities have been scaled by 20 times (see text). This composite image was generated from two independent simulations including vibrational modes 34–42 and 43–45 (Figs. S14 and S15); the vertical dashed line indicates where the two data sets are joined. The colour bars show fractional changes in relative transmittance (normalised absorbance) due to the magnetic field.

IR absorption, for instance around 200 cm$^{-1}$. To gain qualitative understanding of this pattern we first develop a simple toy model before moving onto a full ab initio analysis of the spectrum.

FIRMS maps have been expertly modelled by Atanasov and Neese[29], among others[15–17], and we follow a similar conceptual approach. Our simple toy model consists of two effective $S = 1/2$ KDs separated by $\Delta$, coupled to a single vibrational mode of energy $\hbar\omega$, for which we consider only the ground

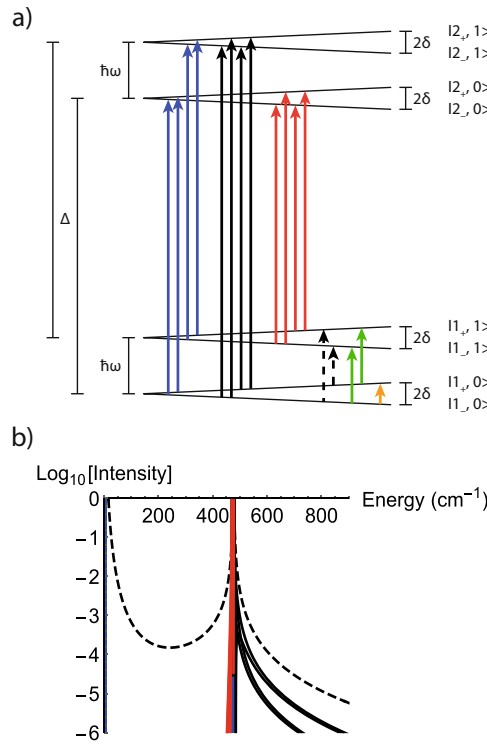

**Fig. 4 Vibronic states and theoretical FIRMS intensity for toy model.**
**a** States of toy model; note that the vibrational energy $\hbar\omega$ is a variable, and all values are considered in (b). Purely electronic transitions are shown in blue, purely vibrational transitions in green, cold vibronic inter-KD transitions in black, vibrationally hot vibronic transitions in red, and cold vibronic intra-KD transitions in dashed black; EPR transitions (not studied in this work) in orange.
**b** Absorption intensity for cold intra-KD vibronic transitions (black dashed lines), cold inter-KD vibronic transitions (solid black lines), hot inter-KD vibronic transitions (red solid lines), and purely electronic (solid blue lines, obscured by the red ones) transitions, under irradiation from an IR source with uniform intensity and uniform vibronic coupling. Purely vibrational transitions are not shown. Constructed with model parameters $F = G = 1$ cm$^{-1}$, $A_v = 10^2$, $A_e = 1$, $\Delta = 474$ cm$^{-1}$ and $\delta = 2$ cm$^{-1}$ (field ca. 2 T).

$n = 0$ and first excited $n = 1$ vibrational quantum states. We label the states as $|N_\pm, n\rangle$ where $N$ is the index of the electronic state, $\pm$ represents each state of the KD, and $n$ is the vibrational state. For simplicity, we assume both KDs have the same $g$-values and hence parameterise the effect of the magnetic field as $\delta = g\mu_B B/2$; the states are presented graphically in Fig. 4a. Without vibronic coupling, the zeroth-order Hamiltonian $\hat{H}_0$ of this toy model in the direct product basis $\{|1_-, 0\rangle, |1_-, 1\rangle, |1_+, 0\rangle, |1_+, 1\rangle, |2_-, 0\rangle, |2_-, 1\rangle, |2_+, 0\rangle, |2_+, 1\rangle\}$ is:

$$\hat{H}_0 = \begin{bmatrix} -\delta & 0 & 0 & 0 & 0 & 0 & 0 & 0 \\ 0 & -\delta+\hbar\omega & 0 & 0 & 0 & 0 & 0 & 0 \\ 0 & 0 & \delta & 0 & 0 & 0 & 0 & 0 \\ 0 & 0 & 0 & \delta+\hbar\omega & 0 & 0 & 0 & 0 \\ 0 & 0 & 0 & 0 & \Delta-\delta & 0 & 0 & 0 \\ 0 & 0 & 0 & 0 & 0 & \Delta-\delta+\hbar\omega & 0 & 0 \\ 0 & 0 & 0 & 0 & 0 & 0 & \Delta+\delta & 0 \\ 0 & 0 & 0 & 0 & 0 & 0 & 0 & \Delta+\delta+\hbar\omega \end{bmatrix} \quad (1)$$

Vibronic transitions are defined as those that involve a change in both the electronic and vibrational states, otherwise the transitions are purely electronic or purely vibrational. There are

two types of vibronic transitions in this toy model: intra-KD transitions (dashed black arrows in Fig. 4a) and inter-KD transitions (solid black and red arrows in Fig. 4a). Due to the low temperature of the experiment (4.2 K) and large CF splitting ($\Delta = 474$ cm$^{-1}$), all absorptions must arise from the four initial states $|1_-, 0\rangle, |1_-, 1\rangle, |1_+, 0\rangle$ or $|1_+, 1\rangle$, and for vibrational modes with $\hbar\omega > 20$ cm$^{-1}$, only cold transitions are relevant (Supplementary Fig. 4). We define the transition matrix elements due to the IR radiation between electronic states as $A_e$ and between vibrational states as $A_v$ (Eq. (S1)). For the system without vibronic coupling described by $\hat{H}_0$, the only possible transitions are purely electronic at $h\nu = 2\delta$, $h\nu = \Delta$ and $h\nu = \Delta \pm 2\delta$ (intensities proportional to $A_e^2$) or purely vibrational at $h\nu = \hbar\omega$ (intensities proportional to $A_v^2$); the observation of multiple transitions in the vicinity of a single electronic excitation in the FIRMS map provides direct evidence for the vibronic coupling. Anticipating our ab initio model (see Supporting Information), we define the vibronic coupling as perturbations to the electronic states in the weak-coupling limit: diagonal terms $G$ express energy shifts and off-diagonal terms $F$ describe coupling between different electronic states (Eq. (S2)). The coupling Hamiltonian $\hat{H}_1$ in the direct product basis is:

$$\hat{H}_1 = \begin{bmatrix} 0 & G & 0 & F & 0 & F & 0 & F \\ G & 0 & F & 0 & F & 0 & F & 0 \\ 0 & F & 0 & G & 0 & F & 0 & F \\ F & 0 & G & 0 & F & 0 & F & 0 \\ 0 & F & 0 & F & 0 & G & 0 & F \\ F & 0 & F & 0 & G & 0 & F & 0 \\ 0 & F & 0 & F & 0 & F & 0 & G \\ F & 0 & F & 0 & F & 0 & G & 0 \end{bmatrix} \quad (2)$$

We can determine the eigenstates of $\hat{H}_0 + \hat{H}_1$ with first-order perturbation theory (Eq. (S3)) and hence plot a theoretical FIRMS spectrum for each class of transition (Fig. 4b). Our theoretical spectrum predicts envelopes of increased intensity for all vibronic transitions around $h\nu \approx \Delta$, in agreement with experiment. For intra-KD vibronic transitions (absorption of an IR photon with $h\nu = 2\delta + \hbar\omega$, where the CF excitation at $\Delta$ is not involved in the transition), we are in the regime where $\delta \ll \Delta$ and hence, for transitions near $h\nu \approx \Delta$, we have that $\hbar\omega \approx \Delta$ and thus $\delta \ll \hbar\omega$. While for inter-KD vibronic transitions (absorption of an IR photon with $h\nu = \Delta + \hbar\omega$ or $h\nu = 2\delta + \Delta + \hbar\omega$, where the CF excitation at $\Delta$ is involved in the transition), we are in the regime where $h\nu \approx \Delta$ and so both $\delta \ll \Delta$ and $\hbar\omega \ll \Delta$. Considering the cold intra-KD vibronic transition, the analytical transition intensity under the approximation that $\delta \ll \Delta$ and $\delta \ll \hbar\omega$ (full expression given in Eq. (S4)), obtained as the square of the off-diagonal transition matrix element, is:

$$I(|1_-, 0\rangle \rightarrow |1_+, 1\rangle) \propto \left( \frac{2F(A_v F\hbar^2\omega^2 - 2A_e\Delta\hbar^2\omega^2 + A_v G(\hbar^2\omega^2 - \Delta^2))}{\hbar^2\omega^2(\hbar^2\omega^2 - \Delta^2)} \right)^2 \quad (3)$$

The intensity of this transition increases when the vibronic coupling ($F$ and $G$), the electronic transition intensity ($A_e$) or the vibrational transition intensity ($A_v$) increase. But notably, the intensity diverges (in first-order perturbation theory) when $\hbar\omega = \Delta$ and, therefore, this toy model predicts that we should expect intense intra-KD vibronic signals when the energy of a vibrational mode is similar to CF gaps in the molecule. Similar expressions occur for the inter-KD vibronic transitions (Eqs. (S5)–(S8)), that diverge when $\delta \rightarrow 0$ and/or $\hbar\omega \rightarrow 0$ (again, in proximity to purely electronic transitions). It appears that these divergences

occur as a consequence of perturbation theory, which may become invalid in cases of degeneracy in the Hamiltonian. Hence, instead of relying solely on perturbation theory, we can exactly diagonalise $\hat{H}_0 + \hat{H}_1$ and calculate the transition intensities (Supplementary Fig. 5a). We observe the same pattern of an envelope of enhanced transition intensity around the electronic transition (and also in our full ab initio model, see below), and hence this effect is not merely an artefact of perturbation theory. This "envelope effect" occurs when the energy of the vibronic excited state ($\hbar\omega \pm 2\delta$, $\hbar\omega + \Delta$ or $\hbar\omega + \Delta \pm 2\delta$) approaches that of the purely electronic excited state ($\Delta$ or $\Delta \pm 2\delta$), but crucially does not require participation of that specific electronic excited state in the transition (e.g. for the intra-KD excitations). Further, the effect can also occur reasonably remotely from the resonance condition, even in the presence of weak coupling (see transitions **A** and **B** in our full analysis below, where $\hbar\omega \pm 2\delta$ is ca. 80 cm$^{-1}$ away from the CF excitation, and yet vibronic coupling is ca. 1 cm$^{-1}$). To explain the physical origin of this "envelope effect", we consider the cold intra-KD vibronic transition $|1_-, 0\rangle' \rightarrow |1_+, 1\rangle'$ at $h\nu = \hbar\omega + 2\delta$. In first-order perturbation theory, the composition of the initial state $|1_-, 0\rangle'$ has no divergences as a function of the vibrational energy ($\hbar\omega$), while the final state $|1_+, 1\rangle'$ takes on a significant admixture of $|2_-, 0\rangle$ and $|2_+, 0\rangle$ when $\hbar\omega$ approaches $\Delta$ (Eq. (S3)), which contributes to the transition probability via the electronic intensity ($A_e$); this effect is maintained when exact diagonalisation is employed as opposed to perturbation theory (Supplementary Fig. 5b and c), but also highlights the fact that the $|1_-, 1\rangle$ state also becomes important, contributing to the transition probability via the vibrational intensity ($A_v$). The presence of an "envelope effect" is a general conclusion that applies to other FIRMS experiments, which have been observed previously but not explained[15], and is rather different than the observation of direct coupling between electronic and vibrational states via avoided crossings[16]. While our toy model focusses on the first CF excitation at ca. 474 cm$^{-1}$ (which leads to intensity in the region 370–550 cm$^{-1}$), we expect enhanced vibronic intensity near the second excited KD at 745 cm$^{-1}$ for the same reasons, which gives rise to the experimentally observed transitions in the range 740–815 cm$^{-1}$ (Supplementary Fig. 16).

**Ab initio FIRMS analysis.** Moving beyond a simple toy model where state energies and vibronic coupling are parameters, we now endeavour to understand the details of the FIRMS map of **1** (Fig. 3a, b) by calculating the vibronic coupling ab initio, as embodied in our FIRMS_sim package[30]. The conceptual framework is similar to the toy model, but now we consider the realistic details of **1**, where the energies of the coupled electronic and vibrational states as a function of magnetic field are obtained from the total Hamiltonian $\hat{H}_T$:

$$
\begin{aligned}
\hat{H}_T &= \hat{H}_{CF} + \hat{H}_{Zee} + \sum_j \left[ \hat{H}_{vib,j} + \hat{H}_{coup,j} \right] \\
&= \sum_{k=2,4,6} \sum_{q=-k}^{k} B_k^q \hat{O}_k^q + \mu_B g_J \vec{B} \cdot \vec{J} + \sum_{j=1}^{3N-6} \left[ \hbar\omega_j \left( n_j + \frac{1}{2} \right) + \hat{H}_{coup,j} \right]
\end{aligned}
\tag{4}
$$

The first and second terms are the electronic CF and Zeeman Hamiltonians, evaluated in the $|m_J\rangle$ basis of the ground $^2F_{7/2}$ spin–orbit multiplet of Yb$^{III}$, the third term is the quantum harmonic oscillator Hamiltonian, evaluated in the basis of vibrational quanta $|n_j\rangle$ for mode $j$, and the fourth term $\hat{H}_{coup,j}$ is the vibronic coupling Hamiltonian; $\mu_B$ is the Bohr magneton, $g_J$ is the Landé $g$-factor for Yb$^{III}$, $\vec{B}$ is the magnetic field vector, $\vec{J}$ is the electronic total angular momentum vector operator, $B_k^q$ are the Stevens CF parameters (CFPs), $\hat{O}_k^q$ are the Stevens operators, $\hbar\omega_j$ is the energy of vibrational mode $j$, $\hbar$ is the reduced Planck constant,

and $N$ is the number of atoms. As $\hat{H}_{CF} + \hat{H}_{Zee}$ commutes with each $\hat{H}_{vib,j}$, and all $\hat{H}_{vib,j}$ commute with one-another in the harmonic approximation, Eq. (4) can be written in the direct product basis $|m_J, n_1, n_2, ...\rangle$ (see Supporting Information). $\hat{H}_{CF}$ is constructed using CFPs from CASSCF-CASPT2-SO calculations (see the "Methods" section; this encodes all information on the CF energies and anisotropic $g$-values) and each $\hbar\omega_j$ is obtained from DFT calculations (see the "Methods" section), where we only consider the $n_j = 0$ and $n_j = 1$ states (thus ignoring vibrational overtones). To construct each $\hat{H}_{coup,j}$, we expand the CFPs for **1** in a Taylor series in the displacement $Q_j$ along normal mode $j$ around equilibrium $Q_j = 0$[31], where $Q_j = 1$ is defined as the zero-point displacement of mode $j$ (see Supporting Information).

$$
\begin{aligned}
B_k^q \left( Q_j, ..., Q_{3N-6} \right) &= B_{k\,eq}^q + \sum_{j}^{3N-6} Q_j \left( \frac{\partial B_k^q}{\partial Q_j} \right)_{eq} \\
&+ \frac{1}{2} \sum_{j}^{3N-6} \sum_{j'}^{3N-6} Q_j Q_{j'} \left( \frac{\partial^2 B_k^q}{\partial Q_j \partial Q_{j'}} \right)_{eq} + ...
\end{aligned}
\tag{5}
$$

$$
\hat{H}_{coup,j} = \sum_{k=2,4,6} \sum_{q=-k}^{k} Q_j \left( \frac{\partial B_k^q}{\partial Q_j} \right)_{eq} \hat{O}_k^q
\tag{6}
$$

In this work, we simulate FIRMS maps by employing a first-order approximation in which the linear term is assumed to be dominant (verified by our ab initio calculations, Supplementary Fig. 6), leading to the vibronic coupling Hamiltonian, Eq. (6). We calculate the vibronic coupling coefficients $\left( \frac{\partial B_k^q}{\partial Q_j} \right)_{eq}$ by performing CASSCF-SO calculations on distorted molecular geometries along each normal mode coordinate and fitting the change in CFP with a third-order polynomial (see Supporting Information)[32]. At equilibrium geometry, the $C_3$ point symmetry of **1** means that only $B_k^q$ with $q = 0, \pm3, \pm6$ are non-zero. However, this constraint can be lost when the molecule vibrates and, thus, up to 27 non-zero $\left( \frac{\partial B_k^q}{\partial Q_j} \right)_{eq}$ contributions to $\hat{H}_{coup,j}$ are possible. Hence, we define the overall vibronic coupling strength for each mode as $S_j$ (Eq. (7))[32,33]; note here that $\mathscr{B}_{k,q}$ are CFPs in Wybourne notation and are linear combinations of the CFPs in Stevens notation $B_k^q$[34].

$$
S_j = \sqrt{\frac{1}{3} \sum_k \frac{1}{2k+1} \sum_{q=-k}^{k} \left| \left( \frac{\partial \mathscr{B}_{k,q}}{\partial Q_j} \right)_{eq} \right|^2}
\tag{7}
$$

To calculate the matrix representation of Eq. (6), we assume the weak-coupling limit where vibrational modes are unaffected by coupling to electronic states, and thus the electronic and vibrational components can be separated, Eqs. (8)–(10) (see Eq. (S9) for the more general expression for multiple modes, and Eqs. (S13)–(15) for the definition of the matrix elements in Eq. (10)).

$$
\hat{H}_{coup,j} = \hat{H}_{coup-e,j} \otimes \hat{H}_{coup-v,j}
\tag{8}
$$

$$
\hat{H}_{coup-e,j} = \left\langle m_{J'} \left| \sum_{k=2,4,6} \sum_{q=-k}^{k} \left( \frac{\partial B_k^q}{\partial Q_j} \right)_{eq} \hat{O}_k^q \right| m_J \right\rangle
\tag{9}
$$

$$
\hat{H}_{coup-v,j} = \left\langle n_j \pm 1 \left| Q_j \right| n_j \right\rangle
\tag{10}
$$

To determine the intensity of the FIRMS transitions driven by the IR radiation, we define the matrix of Einstein coefficients $B_j$ in

the direct product basis, Eqs. (11)–(13) (which can be generalised in the same manner as Eq. (8) is by Eq. (S9)). Here, the first term corresponds to the contribution of magnetic dipole transitions and the second term to electric dipole transitions, where $\varepsilon_0$ and $\mu_0$ are the permittivity and permeability of free space, and $\hat{\mu}_M$ and $\hat{\mu}_E$ are the magnetic and electric dipole moment operators, respectively. In this work we assume that purely vibrational transitions are electric dipole in nature, while purely electronic transitions between the $m_J$ states of the $J = 7/2$ manifold are magnetic dipole in nature; the latter is justified because these are intra-configurational $4f$–$4f$ transitions with little change in $4f$ electron density. Calculation of the magnetic dipole matrix elements in the $|J, m_J\rangle$ basis is straight forward with $\hat{\mu}_M = \mu_B g_J \vec{\hat{J}}$, however the electric dipole matrix elements are more involved (see Supporting Information). Once the matrix of Einstein coefficients $B$ is determined, we apply a change of basis using the eigenvectors of $\hat{H}_T$ to obtain $B$ in the coupled basis.

$$B = B_e \otimes B_v \tag{11}$$

$$B_e = \frac{2\pi\mu_0}{3\hbar^2 c^2} \hat{\mu}_M^\dagger \hat{\mu}_M \tag{12}$$

$$B_v = \frac{2\pi}{3\hbar^2 c^2 \varepsilon_0} \hat{\mu}_E^\dagger \hat{\mu}_E \tag{13}$$

To construct a single FTIR absorption spectrum at fixed magnetic field, we assume that each transition is described by a normalised Gaussian function centred at the transition energy $\Delta E$ with an empirical full-width-at-half-maximum of 5 cm$^{-1}$, and peak height equal to the product of the corresponding Einstein coefficient and the Boltzmann population of the initial state. To simulate a FIRMS map we must calculate multiple absorption spectra as a function of magnetic field. However, as the experiments are performed for powder samples, we must integrate over the orientation of the external magnetic field in $\hat{H}_{Zee}$; here we use the Zaremba–Conroy–Wolfberg (ZCW) model with an integration level of 3 (89 directions evenly distributed on a hemisphere) for all calculations[35]. Furthermore, we must account for the unpolarised nature of the incident IR radiation, and follow the treatment of Nehrkorn et al. to integrate all possible polarisation vectors of the IR radiation in the experimental Voight configuration[36]. Finally, we calculate the FIRMS map by performing exactly the same normalisation procedure as for the experimental map (see the "Methods" section).

Using this ab initio approach, we simulate a FIRMS map for the electronic states of **1** coupled to selections of vibrational modes; calculation of the full vibronic manifold with all vibrational modes is far beyond current computational feasibility, and is not necessary as only modes in energetic proximity of one another need be modelled simultaneously. When considering more than one vibrational mode, $\hat{H}_{\text{coup},j}$ does not include direct coupling of both modes (i.e., the $\left(\frac{\partial^2 B_k^q}{\partial Q_j \partial Q_{j'}}\right)_{eq}$ term of Eq. (5) is not included in Eq. (6)), but Eq. (4) does allow vibrational mode interactions via the electronic states.

Looking now at the experimental data, we focus on the most intense signals near the first electronic transition (370–550 cm$^{-1}$, Fig. 3). However, there are also strong signals in proximity to the second electronic transition (775 cm$^{-1}$) and far weaker signals at 167, 238, 553 and 581 cm$^{-1}$, all of which are more distant from any electronic transitions; these are discussed in the Supporting Information, along with a full assignment of the proceeding signals. Examining the 370–550 cm$^{-1}$ region, we can identify intense field-dependent signals emerging from 393, 407, 444, 468,

474, and 520 cm$^{-1}$ in zero-field (**A**–**F**, respectively, Figs. 3b and 5). Signals below the first CF state (i.e. 370–474 cm$^{-1}$, **A**–**C**) must mainly be due to cold intra-KD vibronic transitions ($|1_\pm, 0\rangle \rightarrow |1_\mp, 1\rangle$) (Fig. 4), because hot transitions are very unlikely at 4.2 K, Supplementary Fig. 4), where the observed energy in the FIRMS map is close to the vibrational energy (as $\delta \ll \hbar\omega$ and hence $h\nu = \hbar\omega \pm 2\delta \approx \hbar\omega$). Signals at or above the electronic excitation (i.e. 474–550 cm$^{-1}$, **D**–**F**) can either be intra- or inter-KD vibronic transitions ($|1_\pm, 0\rangle \rightarrow |1_\mp, 1\rangle$, $|1_\pm, n\rangle \rightarrow |2_\pm, n'\rangle$ or $|1_\pm, n\rangle \rightarrow |2_\mp, n'\rangle$) or purely electronic transitions ($|1_\pm, n\rangle \rightarrow |2_\pm, n\rangle$, Fig. 4). Signals that move to higher energies with increasing field are electronically cold, originating from $|1_-, n\rangle$, whilst those moving to lower energies are electronically hot, originating from $|1_+, n\rangle$. On this basis, and considering the DFT-calculated vibrational spectrum, we can assign signals **A** and **B** (corresponding to peak 2a in the luminescence spectrum, Fig. 2) as intra-KD transitions coupled to vibrational modes $j = 34$–36 (Supplementary Fig. 8, Supplementary Movies 13–15), and signal **C** as an intra-KD band coupled to modes $j = 37$–39 (Supplementary Fig. 9, Supplementary Movies 16–18). Signals **D** and **E** are complicated as they contain contributions from purely electronic and intra-KD bands coupled to modes 40–42 (Supplementary Fig. 10, Supplementary Movies 19–21); we rule out inter-KD hot bands (requiring $\hbar\omega < 20$ cm$^{-1}$) arising from acoustic phonon modes as the vibrational transition intensities will be negligible compared to intramolecular modes. The intense signal **F** (corresponding to peak 2b in the luminescence spectrum, Fig. 2) could either be an intra-KD or an inter-KD vibronic transition, or both; the former would involve either modes 43–45 (Supplementary Fig. 11, Supplementary Movies 22–24) and the latter would involve modes 4 and 5 (Supplementary Fig. 12, Supplementary Movies 1 and 2).

Using our fully ab initio method for calculation of FIRMS maps (see above and Supporting Information), we see good agreement for each of the individual bands **A**–**E** in separate simulations of each mode for those discussed above (Supplementary Figs. 8–10). Signal **F**, however, could have two origins, and in the light of our simulations, we believe that **F** is an intra-KD transition coupled to modes 43–45 (Supplementary Fig. 11); simulations for an inter-KD transition coupled to modes 4–5 show almost no intensity in a FIRMS map in this region

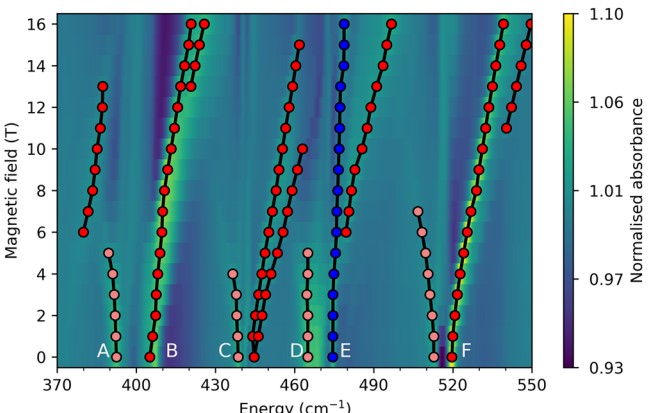

**Fig. 5 Experimental FIRMS map for 1 with transitions highlighted.** Experimental FIRMS map in the range 370–550 cm$^{-1}$, with purely electronic transitions shown in blue, and vibronic intra-KD transitions in red/pink (cold/hot). Assignments are based on ab initio simulation (Fig. 3c). Note that the weak field-independent signals (vertical lines) may be pure vibrational modes, but as they appear at the turning points of the raw transmission spectra they could very well be instrumental artefacts.

(Supplementary Fig. 12). To simulate the entire spectrum, we would thus need to include all 12 modes ($j = 34$–$45$), but unfortunately this is computationally unfeasible. Thus, we perform a 9-mode simulation for signals **A**–**E** ($j = 34$–$42$) and consider **F** ($j = 43$–$45$) in a separate simulation. Using the fully ab initio method, we observe excellent reproduction of the signal positions but there is poor agreement with the experimental intensities (Supplementary Figs. 11 and 13); this is due to the gas-phase DFT-calculated IR vibrational transition intensities (where there is no representation of the molecular environment) that are only in modest agreement with experiment (Supplementary Figs. 2 and 3; although note that the vibrational energies are predicted remarkably well). We can remedy this by scaling our vibrational transition intensities such that their relative values match those of the experimental zero-field infrared spectrum (Supplementary Fig. 3; note that calculated intensities are proportional to the integral of the IR band). However, this only considers the relative intensity of the vibrational bands and it is the absolute intensity that enters the Einstein coefficients in Eqs. (12) and (13). Hence, after scaling the relative vibrational intensities to experimental data, we are left with a free parameter correcting the relative magnetic dipole transition intensity (transitions between the electronic states) to the electric dipole intensity of the vibrations; this is a simple scalar introduced to the right-hand-side of Eq. (12). Exploring magnetic dipole scaling factors in the range $10^{-3}$–$10^3$, we find best agreement with experiment with a factor of 20 (Supplementary Figs. 14 and 15). To compare the full FIRMS map with experiment in the 370–550 cm$^{-1}$ range, we build a composite (summative) image of our ab initio simulations of signals **A**–**E** (modes 34–42) and signal **F** (modes 43–45) (Fig. 3c). Overall the map shows good agreement with the experiment, where the shapes and relative intensities of the features within each signal are consistent with the experimental data. Of course, these simulations are subject to the errors of the DFT-calculated vibrational modes, for which the error in energy is most significant for signals **A** and **B** (Supplementary Fig. 2), and the error in intensity affects all modes (Supplementary Figs. 2 and 3); we have attempted to correct the latter by scaling the zero-field IR intensities, but this is not an ideal solution. In the simulated FIRMS map we clearly see that the positions of signals **A** and **B** are shifted compared to the experiment, and that the intensity of signals **C** and **F** are too great and too small, respectively, compared to the experiment (Fig. 3). These differences are laid-bare in cut plots at fixed fields which do not show good agreement with experiment due to these DFT-based errors (Supplementary Fig. 17). One may be curious if agreement could be improved by instead scaling the ab initio-calculated vibronic coupling, $\hat{H}_{\text{coup}}$. Applying a uniform scaling factor to all modes results in a variety of changes in the simulated FIRMS map (Supplementary Fig. 18). When the coupling is increased by a factor of 2, signals **A** and **B** move closer to their experimental positions while signal **E** shifts far from its experimental value. Decreasing the coupling has a much more detrimental effect, as signals **A** and **B** move closer together, and all vibronic signals are much less intense. Therefore, it is our belief that any attempt to correct the simulation further by tweaking either absolute vibrational energies, intensities, or vibronic coupling parameters, defeats the purpose of an ab initio simulation. Instead, we argue that the profile of each signal, which arises from the ab initio-calculated vibronic coupling, is in good agreement with the experimental data.

While the FIRMS map allows us to observe vibronic transitions, we have used a "toy model" to show that such spectra are subject to an "envelope effect", making them most sensitive to vibronic transitions near electronic excited states

(Fig. 4b, Eq.(3) and Supplementary Eqs. (4)–(8)). We can now demonstrate this effect further using our full ab initio method by shifting the energy of the purely electronic transition ($|1_\pm, 0\rangle \rightarrow |2_\pm, 0\rangle$) in the 370–520 cm$^{-1}$ region and simulating FIRMS maps for modes 34–36 (signals **A** and **B**). When the electronic transition is placed at 370 cm$^{-1}$ the vibronic signals are clear (Supplementary Fig. 18a) and when it is shifted to 400 cm$^{-1}$ (on top of the vibrational modes at 405–413 cm$^{-1}$) the intensity is drastically increased (Supplementary Fig. 18b, along with the separation of the signals due to the vibronic coupling). When the electronic transition is moved away to 420 cm$^{-1}$ a similar FIRMS intensity is observed for **A** and **B** as it was in the 370 cm$^{-1}$ simulation (Supplementary Fig. 18c cf. Supplementary Fig. 18a, because the distance from the vibrational mode energies is similar ~30 cm$^{-1}$ and the envelope effect is roughly symmetrical, Fig. 4b), and the FIRMS intensity decreases consistently when it is moved further away to higher energy up to 520 cm$^{-1}$ (Supplementary Fig. 19). Thus, owing to the envelope effect, FIRMS maps do not provide a direct measure of the strength of the vibronic coupling for all modes. However, our high-quality modelling of the FIRMS map here serves as a detailed benchmark of our ab initio calculation of the vibronic coupling and, thus, we can examine the coupling strength for all vibrational modes. The values of $S$ for all vibrational modes (Fig. 6 and Supplementary Fig. 22, Supplementary Table 6) reveal that those in the 370–550 cm$^{-1}$ region are not more strongly coupled than modes at other energies. While modes 35 and 36 (responsible for signal **B**) have the second-largest vibronic coupling overall, they have similar values to modes 26 and 27 at 305 cm$^{-1}$, but we do not see any intra-KD transitions around 300 cm$^{-1}$ in the FIRMS experiment; this is due to the envelope effect. Overall, there are six pairs of modes with $S > 0.75$ cm$^{-1}$ (7, 8, 14, 15, 22, 23, 26, 27, 35, 36, 142 and 143); interestingly, these modes all have E-symmetry, which breaks the $C_3$ point symmetry of the equilibrium structure (see Supplementary Movies). However, it is not as simple as A- vs. E-symmetry dictating the strength of vibronic coupling: the modes listed above all involve significant distortions to the first coordination sphere of Yb$^{\text{III}}$ while, for instance, there are E-symmetry modes at 477 and 762 cm$^{-1}$ (modes 41–42 and 58–59, respectively) that involve peripheral motion (see Supplementary Movies) and hence have small $S$ values of 0.12 and 0.02 cm$^{-1}$, respectively. However, there are also A-symmetry modes that involve distortions to the first coordination sphere which have significantly weaker coupling than the modes that couple most strongly (e.g. mode 34 at 406 cm$^{-1}$ with $S = 0.58$ cm$^{-1}$). Therefore, we can conclude that modes that distort the first coordination sphere and break the local point symmetry have the strongest vibronic coupling. The first conclusion is intuitive and has been remarked upon by other authors recently, both for SMMs and molecular qubits[11,37–42], but crucially here this coincides with direct measurement of the vibronic coupling in tandem with high-quality ab initio calculation of the same. Further corroborating our analysis, we find that there is excellent agreement between the strongly coupled modes below 300 cm$^{-1}$ and peaks 1a–1d in the luminescence spectrum (Fig. 2 and Supplementary Fig. 23). Despite having very strong vibronic coupling, these features are weak in the FIRMS map due to the envelope effect (although can be observed, Supplementary Fig. 20), but appear in the luminescence spectrum as this is a spontaneous emission experiment compared to transitions driven by IR photons in FIRMS; the only outlier is the absence of modes 7 and 8 (99 cm$^{-1}$, $S = 0.83$ cm$^{-1}$) in the luminescence spectrum which we cannot presently explain.

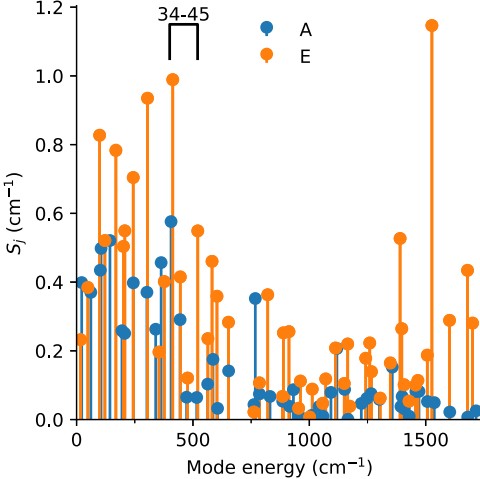

**Fig. 6 Ab initio calculated vibronic coupling strength.** $S_j$ of the vibrational modes of $\mathbf{1_{opt}}$ with A (blue) and E (orange) symmetry. Modes 34–45 are highlighted.

In summary, we have measured the FIRMS map for [Yb(trensal)] (**1**) and developed an ab initio model to calculate the vibronic coupling and hence simulate the map. Our theoretical model shows that vibronic transitions in FIRMS experiments are subject to an "envelope effect" and thus should be most intense near electronic excitations, explaining the structure of our spectra; this is a general phenomenon that has not been described before. Our fully ab initio calculation of the FIRMS map shows excellent agreement with experiment and thus directly validates our approach for calculating vibronic coupling. Hence, we can determine the vibrational modes which are most strongly coupled to the electronic states; for [Yb(trensal)] these are E-symmetry modes involving significant motion in the first coordination sphere at 99, 169, 243, 305, 413 and 1527 cm$^{-1}$. To extract yet more information on the vibronic coupling, future low energy (20–100 cm$^{-1}$) and single crystal FIRMS experiments will allow direct probing of acoustic phonon modes and anisotropy effects, respectively, which are both crucial in low-temperature Raman relaxation of SMMs and decoherence in spin qubits. Indeed, we can use the same computational methodology to predict magnetic relaxation due to vibronic coupling in single-molecule magnets[32], and similar methods can be used to directly probe the contribution of vibronic coupling to decoherence in molecular qubits[7]. Only by combined experimental and theoretical studies such as these can we unravel the details of vibronic coupling in molecules and thus begin to develop guidelines for control of this crucial interaction.

## Methods

**FIRMS measurements.** FIRMS measurements were made on 5 mg of a poly-crystalline sample of **1** at 4.2 K for IR energies <900 cm$^{-1}$ using a Bruker Vertex 80v vacuum FTIR spectrometer with a resolution of 0.3 cm$^{-1}$. FTIR spectra were recorded under a series of applied magnetic fields from 0 to 16 T in the Voigt geometry such that propagation of the incident radiation was perpendicular to the applied field. Transmission was detected using a Si bolometer placed immediately behind the sample (in the magnetic field) in order to minimise loss of power. Transmitted intensity spectra were measured in 1 T field steps. Here, the strong field-independent dips in transmission are due to a combination of electric-dipole-active vibrational absorptions and an instrumental function caused by standing waves in the far-IR propagation system (Supplementary Fig. 7a). To improve the signal-to-noise ratio, each FTIR measurement was repeated four times at each field step, then averaged. To distinguish field-dependent excitations from those that are field independent, spectra at each magnetic field step were divided by the average of all spectra, resulting in clear "magnetic" spectral features above a more-or-less flat baseline and successful suppression of strong field-independent 'dips' in trans-mittance (Fig. 3, Supplementary Fig. 7b). This normalisation procedure does,

however, introduce artefacts wherever the raw transmission is near zero (e.g. around 0 and 720 cm$^{-1}$, Supplementary Fig. 7) due to division of zero-by-zero; these 'blind spots' are due to destructive interference in the beam splitter employed in the FTIR spectrometer. We note also that, outside of these blind spots, a few weak field independent signals remain after background division, which could also be instrumental artefacts as they occur at the turning points of the raw trans-mission spectra. As the measurements were performed on a polycrystalline sample, all molecular orientations in the FIRMS map are sampled at once which, in turn, results in a continuous magnetic field dependent absorption profile superimposed onto the raw FTIR spectrum.

**Ab initio calculations.** Geometry optimisation and calculation of the normal modes of vibration of **1** was performed in the gas-phase using unrestricted DFT within the Gaussian 09 rev. D package[43]. The X-ray crystallographic structure was used as a starting point, and all atomic positions were optimised simultaneously. The PBE0 density-functional was used in conjunction with Grimme's D3 disper-sion correction[44–46], the cc-pVDZ basis set was used for carbon and hydrogen atoms and the cc-pVTZ basis set was used for nitrogen and oxygen atoms[47,48], while the Stuttgart RSC 1997 effective core potential (ECP) was employed for the 28 core electrons of ytterbium and the remaining valence electrons were described with the corresponding valence basis set[49–51]. Symmetry was enabled in the optimisation to preserve the C$_3$ point group.

We use OpenMolcas to perform CASSCF-(CASPT2)-SO calculations for the crystallographic, optimised and distorted geometries of **1**[52]. Basis sets from the ANO-RCC library were employed with VTZP quality for Yb, VDZP quality for the N atoms and O atoms, and VDZ quality for all remaining atoms[53,54]. Density fitting of the two-electron integrals using the acCD scheme was performed to speed up the calculations for the distorted structures, while a Cholesky decomposition was used for all other calculations with a threshold of 10$^{-8}$ [55]. The active space consisted of 13, 4$f$ electrons in the seven 4$f$ orbitals of Yb$^{III}$. State-averaged CASSCF calculations were performed for seven roots of the $S = 1/2$ state and then mixed by spin–orbit coupling using the RASSI module[56]. For the crystallographic and optimised structures, single state CASPT2 corrections to the energies of the seven $S = 1/2$ roots were calculated using the NOMULT keyword, prior to RASSI[57]. SINGLE_ANISO was used to decompose the spin–orbit wave functions into the CF Hamiltonian formalism, using a fixed reference frame determined from the optimised structure[58]. Here we report the CF parameters in the context of the Stevens operator equivalent formalism (Supplementary Table 9)[59].

## Data availability

Raw research data files supporting this publication (including computational input/output files, and raw spectral data) are available at https://doi.org/10.48420/16698526.

## Code availability

The code used to generate the ab initio simulations "FIRMS_sim" is available at https://gitlab.com/chilton-group/firms_sim/-/releases/V1.1

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

## Acknowledgements

We thank The University of Manchester for access to the Computational Shared Facility, the EPSRC (studentship to J.G.C.K.), and The Royal Society (University Research Fellowship to N.F.C.). This project has received funding from the European Research Council (ERC) under the European Union's Horizon 2020 research and innovation programme (grant agreement No. 851504 to N.F.C.) and the US Department of Energy (under DE-SC0020260 to S.H.). A portion of this work was performed at the NHMFL which is supported by the US National Science Foundation (grant number DMR-1644779) and the State of Florida. S.P. thanks the VILLUM FONDEN for research grant 13376. We thank Prof. Richard Winpenny, Prof. Eric McInnes, Prof. David Collison and Prof. David Mills for insightful comments.

## Author contributions

J.M. and S.H. designed the experiment. C.D.B. synthesised and purified the compound. S.P. performed and interpreted the luminescence experiments. J.M., J.N. and M.O. performed the FIRMS measurements. J.G.C.K. and N.F.C. developed theory, wrote code, performed calculations and interpreted spectra. J.M., J.G.C.K., S.H. and N.F.C. wrote the manuscript with contributions from all authors.

## Competing interests

The authors declare no competing interests.
