## [Peer Review File · Nature Communications]

Reviewers' Comments:

Reviewer #1:

Remarks to the Author:

Kragoskow et al have now presented a revised version of their manuscript entitled "Analysis of vibronic coupling in a 4f molecular magnet with FIRMS". The main conclusion of the work has remained unchanged: ab initio simulations are able to offer insights into vibronic coupling and predict the main features of FIRMS experiments. The authors have taken on board all the main criticisms of the last round of review and have improved the manuscript substantially.

I support the publication of this manuscript in Nature Communications provided that the following points are addressed:

1) The authors claim that the comparison between simulations and experiments validate the computational strategy for the prediction of vibronic coupling, which can be used for further analysis. How much does the simulation of FIRMS vary with respect to the computed spin-phonon coupling coefficients? I suggest the authors add a few plots as supplementary information where they rescale vibronic coupling and show how results depend on it.

2) Is there a correlation between predicted IR intensity and the value of vibronic coupling in simulating FIRMS? In other words, can the authors match the experimental maps by rescaling vibronic coupling instead of rescaling IR intensities?

3) The authors discuss the relation between spin-phonon coupling in 4f compounds and its dependence on symmetry. I believe these recent works should be referenced in the manuscript

1) J. Am. Chem. Soc. 2021, 143, 42, 17305–17315

2) J. Am. Chem. Soc. 2021, 143, 34, 13633–13645

Reviewer #2:

Remarks to the Author:

In their revised version, Kragoskow et al. largely improved the manuscript as compared to the original version, and took into account the remarks and proposals of the two referees.

This work presents a combined analysis of the vibronic in a well known Yb(III) complex by FIRMS, a toy model and ab initio calculations.

More specifically, the authors show that there is an envelope effect around the electronic transition due to the vibronic coupling.

All the observed bands of the FIRMS are assigned thanks to the ab initio support.

It is furthermore shown that the vibrational modes the most active for vibronic coupling are those involving atoms of the coordination sphere with a E-symmetry.

This work develops a solid ab initio tool for the ab initio description of the vibronic coupling in lanthanide complexes

which allows to assign with confidence the FIRMS bands and quantify the vibronic coupling.

For those reasons, I consider that this article deserves to be published in Nature Communications.

The last column of matrix S3 does not appear correctly in the pdf.

We thank the reviewers for their time in assessing the manuscript, and respond to their comments below in RED.

Reviewer #1 (Remarks to the Author):

Kragoskow et al have now presented a revised version of their manuscript entitled "Analysis of vibronic coupling in a 4f molecular magnet with FIRMS". The main conclusion of the work has remained unchanged: ab initio simulations are able to offer insights into vibronic coupling and predict the main features of FIRMS experiments. The authors have taken on board all the main criticisms of the last round of review and have improved the manuscript substantially.

I support the publication of this manuscript in Nature Communications provided that the following points are addressed:

Thank you.

1) The authors claim that the comparison between simulations and experiments validate the computational strategy for the prediction of vibronic coupling, which can be used for further analysis. How much does the simulation of FIRMS vary with respect to the computed spin-phonon coupling coefficients? I suggest the authors add a few plots as supplementary information where they rescale vibronic coupling and show how results depend on it.

2) Is there a correlation between predicted IR intensity and the value of vibronic coupling in simulating FIRMS? In other words, can the authors match the experimental maps by rescaling vibronic coupling instead of rescaling IR intensities?

Scaling the vibronic coupling coefficients has a drastic effect on the profile of the spectrum, not just intensities. So to answer point 2, no, this does not work. To address points 1 and 2, we have added Figure S18 to show the effect, and discussed briefly in the manuscript. Given that, overall, the structure of our calculated spectrum is quite a good match to the experimental one, we contend that the ab initio calculations have made a reliable estimate of the vibronic coupling strength.

3) The authors discuss the relation between spin-phonon coupling in 4f compounds and its dependence on symmetry. I believe these recent works should be referenced in the manuscript

1) J. Am. Chem. Soc. 2021, 143, 42, 17305–17315

2) J. Am. Chem. Soc. 2021, 143, 34, 13633–13645

Done.

Reviewer #2 (Remarks to the Author):

In their revised version, Kragoskow et al. largely improved the manuscript as compared to the original version, and took into account the remarks and proposals of the two referees.

This work presents a combined analysis of the vibronic in a well known Yb(III) complex by FIRMS, a toy model and ab initio calculations.

More specifically, the authors show that there is an envelope effect around the electronic transition due to the vibronic coupling.

All the observed bands of the FIRMS are assigned thanks to the ab initio support.

It is furthermore shown that the vibrational modes the most active for vibronic coupling are those involving atoms of the coordination sphere with a E-symmetry.

This work develops a solid ab initio tool for the ab initio description of the vibronic coupling in lanthanide complexes

which allows to assign with confidence the FIRMS bands and quantify the vibronic coupling.

For those reasons, I consider that this article deserves to be published in Nature Communications.

Thank you.

The last column of matrix S3 does not appear correctly in the pdf.

We hope this has finally been fixed.